# EBV Reactivation and Disease in Allogeneic Hematopoietic Stem Cell Transplant (HSCT) Recipients and Its Impact on HSCT Outcomes

**DOI:** 10.3390/v16081294

**Published:** 2024-08-14

**Authors:** Nancy Law, Cathy Logan, Randy Taplitz

**Affiliations:** 1Division of Infectious Diseases and Global Public Health, Department of Medicine, University of California, La Jolla, CA 92093, USA; 2Division of Infectious Diseases, Department of Medicine, City of Hope National Medical Center, Duarte, CA 91010, USA; rtaplitz@coh.org

**Keywords:** Epstein–Barr virus, allogeneic transplant, post-transplant lymphoproliferative disorder, EBV-specific cytotoxic T-cells

## Abstract

The acquisition or reactivation of Epstein–Barr virus (EBV) after allogeneic Hematopoietic Stem Cell Transplant (HSCT) can be associated with complications including the development of post-transplant lymphoproliferative disorder (PTLD), which is associated with significant morbidity and mortality. A number of risk factors for PTLD have been defined, including T-cell depletion, and approaches to monitoring EBV, especially in high-risk patients, with the use of preemptive therapy upon viral activation have been described. Newer therapies for the preemption or treatment of PTLD, such as EBV-specific cytotoxic T-cells, hold promise. Further studies to help define risks, diagnosis, and treatment of EBV-related complications are needed in this at-risk population.

## 1. Introduction and Epidemiology

Epstein–Barr virus (EBV) is a ubiquitous human herpesvirus that is largely acquired in childhood, leading to a seroprevalence in adults worldwide of approximately 90% [1]. EBV escapes host innate and adaptive immune responses and establishes life-long latent infection in B-lymphocytes. Acute infection is frequently subclinical, but can cause mononucleosis; EBV infection may also be associated with a number of different human malignancies as well as autoimmune disorders. EBV is also the most common virus associated with secondary hemophagocytic lymphohistiocytosis (HLH), which can be a life-threatening hyperinflammatory syndrome [2].

One of the most vulnerable populations affected by EBV are hematopoietic stem cell transplantation (HSCT) recipients. Either primary or reactivation EBV in an immunocompromised transplant patient can be associated with post-transplant lymphoproliferative disease (PTLD). This proliferative B-cell disorder is associated with loss of cytotoxic T-cell control and unchecked B-cell activation. While rates of primary or reactivation EBV and PTLD vary based on transplant type, reported EBV reactivation/infection rates are as high as 30% in allogeneic HSCT (allo-HSCT), and PTLD incidence ranges anywhere between 1 and 10% [1,3,4,5]. 

PTLD usually occurs 60 to 90 days after HSCT. The majority of cases will occur between 1 and 6 months after HSCT [5,6]. There are no documented cases occurring before 1 month and it is rare to occur after 1 year [5,6]. EBV-related PTLD remains a life-threatening complication, associated with a high mortality rate of 50 to 90% [4,6,7].

In patients with EBV reactivation, the primary mechanisms of death include severe immune dysregulation, such as cytokine storms or hemophagocytic lymphohistiocytosis (HLH), leading to systemic inflammation and multi-organ failure [5,8,9,10]. Direct viral effects, including significant tissue damage and organ invasion, also contribute to mortality [9]. In contrast, patients with EBV-positive post-transplant lymphoproliferative disorder (PTLD) primarily die from the uncontrolled proliferation of EBV-infected B-cells, resulting in extensive organ infiltration and dysfunction [8,9]. The immunosuppressive therapy required for transplant patients exacerbates this by hindering the immune response, allowing unchecked lymphoproliferation and leading to complications such as cytokine release syndrome and severe infections [5,8,9,10]

EBV-associated PTLD spans a spectrum of subtypes from benign polyclonal proliferations to malignant monoclonal lymphomas. PTLD includes several subtypes with varying outcomes, including reactive hyperplasia, diffuse large B-cell lymphoma (DLBCL), and plasmablastic lymphoma. Reactive hyperplasia is a benign condition, characterized by the proliferation of normal lymphoid tissue in response to an antigenic stimulus, often linked to EBV, typically with a favorable prognosis. It is an early lesion in PTLD and usually does not progress to malignancy but may be resolved with reduced immunosuppression [11,12].

Polymorphic PTLD represents a transitional phase with polymorphic cellular infiltrate, architectural distortion of lymphoid tissue, and the presence of EBV in most cases. These lesions show clonal or oligoclonal lymphoid populations but do not meet the criteria for a specific lymphoma type [11,13,14]. They involve malignant, monoclonal B-cell proliferation. Prognosis varies based on the stage at diagnosis, response to immunosuppression reduction, and treatment efficacy. Intensive treatment is often required, with outcomes ranging from favorable to poor [11,12,13,14]

Monomorphic PTLD includes B-cell neoplasms such as diffuse large B-cell lymphoma (DLBCL), Burkitt lymphoma, and plasma cell myeloma. These are monoclonal, aggressive, and resemble their counterparts in immunocompetent hosts. T/NK-cell neoplasms are less common and include peripheral T-cell lymphoma and hepatosplenic T-cell lymphoma. These lesions have a poorer prognosis compared to B-cell PTLD [11,13,14].

Plasmablastic lymphoma is a rare, highly aggressive subtype of PTLD, often involving the oral cavity and strongly associated with EBV. It has a poor prognosis due to its aggressive behavior and resistance to conventional treatments. Some patients may respond to high-intensity chemotherapy. Early diagnosis and aggressive treatment are crucial for improving outcomes [11].

Classical Hodgkin lymphoma-type PTLD shares histological features with classical Hodgkin lymphoma and is usually EBV-positive. This type has a better prognosis with appropriate treatment [11].

Imaging modalities such as CT and PET-CT scans are crucial in the initial staging and response assessment of PTLD. CT scans help identify the extent of disease, while PET-CT scans provide functional imaging that can distinguish between active disease and fibrotic tissue. PET-CT is particularly valuable in assessing the metabolic activity of lesions, guiding biopsies, and evaluating treatment response [13,15,16]

A biopsy is essential for diagnosing PTLD. It allows for histopathological examination, immunophenotyping, and molecular studies to classify the PTLD subtype accurately. The biopsy should be sufficiently large to provide material for all necessary tests, including EBV-encoded RNA (EBER) in situ hybridization, immunohistochemistry, and clonality studies [14].

## 2. Risk Factors

While multiple factors have been associated with EBV reactivation, T-cell depletion is one of the major risk factors [17,18] (Table 1). Van Esser et al. retrospectively reviewed and compared EBV-seropositive recipients of a T-cell-depleted (TCD) allo-HSCT with EBV-seropositive recipients of a non-T-cell-depleted allo-HSCT and found that recurrent reactivation was observed more frequently in recipients of a TCD graft, and EBV–lymphoproliferative disease (EBV-LPD) occurred only after TCD-SCT [10].

In attempting to prevent graft-versus-host disease (GVHD) post-transplant, many cancer centers have incorporated the use of anti-thymocyte globulin (ATG) or alemtuzumab; this has been shown to be another risk factor of EBV reactivation [10,19]. Multiple studies have compared ATG with no ATG in allo-HSCT and have shown EBV reactivation ranging from 3.6 to 33% with ATG versus 1.4 to 3% without ATG [20,21,22,23]. These variations are attributed to different formulations and dosing of the ATG used in the studies. Burns et al. conducted a study on 186 adult patients who underwent consecutive allogeneic hematopoietic stem cell transplantation (allo-HSCT) with alemtuzumab T-cell depletion. Their findings indicated a cumulative incidence of EBV reactivation at 48% within one year, with a high-level reactivation incidence of 18%. Additionally, eight patients were diagnosed with PTLD during this period [24].

Other established risk factors for EBV reactivation and PTLD are older age (age > 50), reduced intensity conditioning, EBV serology mismatch, second HSCT, pre-transplant splenectomy, HLA mismatch and haploidentical transplant, infusion of mesenchymal stromal cells, grade II to IV GVHD, and cytomegalovirus (CMV) reactivation [17,25,26,27,28,29,30,31,32]. Landgren et al. also validated some of these characteristics in their study and concluded that these subgroups of patients may benefit from prospective monitoring of EBV reactivation and early intervention [29]. Factors identified that reduce the risk of EBV-related PTLD include rituximab use within 6 months pre-HSCT, sirolimus use for GVHD prophylaxis, CD4+ T-lymphocyte count greater than 50 at day 30, and depletion of both T- and B-cells, rather than T-cells alone [26,29,33,34]. There are mixed data as to whether post-transplant cyclophosphamide (without ATG) increases EBV reactivation or is protective, and further studies will be required to make a definitive conclusion [35].

## 3. The Role of Surveillance

Because EBV viremia often precedes the onset of PTLD by several weeks in high-risk transplant patients, monitoring EBV DNA levels in peripheral blood is used to predict the likelihood of PTLD development. Evidence-based guidelines from the European Conference on Infections in Leukemia (ECIL) recommend initiating EBV screening no later than four weeks post-transplantation and continuing for at least three months in high-risk allogeneic HSCT recipients [28]. High-risk factors include T-cell depletion (either in vivo or ex vivo), EBV serology mismatch, cord blood transplantation, HLA mismatch, acute or chronic GVHD requiring intensive immunosuppressive therapy, high EBV viral load, and splenectomy [36]. For patients without detectable EBV DNA, weekly screening is advised, whereas more frequent sampling is suggested for those with rising EBV DNA levels due to the virus’s rapid doubling time of approximately 56 h [37]. Quantitative NAT assays, typically using PCR amplification, are recommended for monitoring EBV viral load. Upon reaching a specific EBV DNA threshold, reducing immunosuppression and initiating pre-emptive rituximab treatment are advised. Extended monitoring is suggested for patients with poor T-cell reconstitution or severe GVHD, those undergoing haploidentical HSCT, those treated with TCD, and those who experience early EBV reactivation [26,28]

In an international survey, Styczyski et al. evaluated management strategies for EBV infections in European transplant centers [38]. The group found that the majority of centers followed weekly monitoring schedules [38]. Certain medical centers have reported more frequent EBV monitoring for specific patient groups, including those recently discharged from the hospital (within the first 2–4 weeks), patients who have undergone haploidentical hematopoietic cell transplantation (HCT), individuals receiving steroid therapy for graft-versus-host disease (GVHD), and those being treated with ruxolitinib [38]. Studies have documented cases of high-grade B-cell lymphomas developing in patients treated with ruxolitinib post-transplant, highlighting the need for careful monitoring and risk assessment in this population [39,40].

The majority of centers monitored up to day +120 or until cessation of cGVHD treatment. However, some centers continued monitoring until the resolution of immunosuppression measured by the recovery of lymphopenia and CD4 cells [38,41]. In the US, there is guidance for monitoring and treatment strategies in subgroups such as cord transplants, including monitoring for the first 100 days, but no consensus guidelines for all allogeneic transplant recipients [31].

## 4. Thresholds of EBV Viral Loads and the Development of End-Organ Disease

The determination of an EBV threshold value for diagnosing EBV-PTLD or other end-organ EBV diseases in allo-HSCT patients is challenging due to the absence of universal standards for nucleic acid testing. This variability stems from differences in testing methodologies and a lack of standardized protocols [42]. Different assays use serum, whole blood (WB), or peripheral blood mononuclear cells (PBMCs) to measure EBV DNA, making it difficult to generalize interpretations. Due to the lack of standardized PCR assay protocols, the ECIL guidelines do not recommend a specific EBV viremia threshold for initiating pre-emptive therapy [28]. However, numerous studies have defined EBV thresholds for preemptive treatment as 1000 copies/mL and 10,000 copies/mL in serum or plasma samples and 40,000 copies/mL in whole blood [28,38,42,43,44]. It is also important to note that some centers use different thresholds depending on the time from transplantation [38]. The rate at which EBV copies increase can be clinically significant as rising EBV viremia is typically due to the proliferation of EBV-infected memory B-cells in the peripheral blood [28,38].

## 5. Preemptive Therapy for EBV Viremia and Preventive Strategies

Management strategies for the prevention and treatment of PTLD include the reduction of immunosuppression, preemptive rituximab or treatment strategies, radiation therapy, immunochemotherapy, chemotherapy, stem cell transplantation, immunotherapy, and surgical excision of a localized lesion.

Preemptive therapy for EBV involves regular testing for EBV viremia and starting therapy, generally with rituximab, at a given EBV threshold. The use and length of rituximab preemptive therapy vary between centers. Many centers continue until one or two negative results of EBV-DNAemia [38]. Some centers use criteria of decreased viremia below local threshold values for the cessation of preemptive therapy [38]. It is important to know that while rituximab can decrease EBV/PTLD it also delays the reconstitution of B-cell immunity, which can lead to other fatal complications such as cytopenia and other infections. Therefore, rituximab should be accompanied by the monitoring of patients’ immune systems and supportive care as needed [28].

Since rituximab is used for both preemptive therapy and PTLD treatment, it is not surprising that it has also been considered for prophylactic use. Patel et al. examined the cumulative incidence of EBV reactivation and EBV-PTLD in allo-HSCT recipients who underwent T-cell depletion with alemtuzumab, comparing those who received pre-HSCT rituximab to those who did not [45]. Their findings indicated that administering rituximab before allo-HSCT significantly reduced the risk of EBV reactivation and EBV-PTLD, without increasing the incidence of aGVHD or infection [45]. A reduction of EBV-DNAemia and PTLD incidence with rituximab was also confirmed by Dominietto et al. in their large, single-center, retrospective study, which also demonstrated reduced grade II to IV acute GVHD [25,28,38,46]

Antiviral agents have been considered for both prophylaxis and treatment. Unfortunately, there are no efficacy data to support their use. Antivirals (e.g., acyclovir, valaciclovir, ganciclovir, valganciclovir, foscarnet) are not effective for latent EBV due to the lack of EBV thymidine kinase [28]. Cidofovir was shown to have in vitro anti-EBV activity, but this was not confirmed in clinical outcomes [47]. Unfortunately, newer antivirals have not been shown to be beneficial. Letermovir has no anti-EBV activity. Maribavir and brincidofovir were shown to have some anti-EBV activity in vitro but no proven clinical benefit, and brincidofovir is no longer available for use [48,49,50]. Currently, antiviral drugs are not recommended to prevent EBV-PTLD after allo-HSCT [28,34].

## 6. Management

Once a diagnosis of probable or proven EBV-PTLD is made, rituximab and immunosuppression reduction are recommended as first-line treatment [27,28]. Second-line options include adoptive cellular therapy (EBV-CTLs or DLI) and chemotherapy with or without the addition of rituximab [28,42]. There is also significant interest in using immune checkpoint therapies for the treatment of EBV-associated cancers [51].

The use of EBV-specific cytotoxic T lymphocytes (CTLs) is currently under evaluation, though these CTLs are not yet commercially available [28,52,53]. Heslop et al. conducted a study across three centers on patients who received infusions of EBV-specific CTLs to prevent or treat EBV-positive lymphoproliferative disease (LPD) following HSCT [54]. They observed minimal toxicity among the participants. In the group that received CTL prophylaxis, none of the 101 patients developed EBV-positive LPD [54]. Additionally, 11 out of 13 patients treated with CTLs for biopsy-confirmed or probable LPD achieved sustained complete remissions. The study also demonstrated the persistence of functional CTLs for up to nine years [54]. Bonifacius et al. studied individualized T-cell products [55]. They evaluated products from stem cell donors, related third-party donors (TPDs), and unrelated (TPDs) from the allogenic T-cell donor registry [55]. They found that 20/29 of their patients were deemed to have complete clinical response, and no infusion reactions were experienced [55].

Tabelecleucel is the first allogeneic, off-the-shelf, EBV-specific T-cell immunotherapy approved for treating relapsed or refractory EBV-positive post-transplant lymphoproliferative disease [56,57,58]. An ongoing global, multicenter, open-label phase 3 trial (NCT03394365) and expanded-access study (NCT02822495) have demonstrated clinical benefits for patients with relapsed or refractory EBV-positive post-transplant lymphoproliferative disease who have no other approved treatment options [56,57,58]. These studies have not identified the safety concerns typically associated with other adoptive T-cell therapies [56,57,58].

Many oncology providers had high expectations for posoleucel, a multivirus-specific T-cell therapy designed for off-the-shelf use against six common viral infections in allo-HCT recipients: adenovirus, BK virus (BKV), cytomegalovirus, Epstein–Barr virus, human herpes virus-6, and JC virus [59,60]. In phase II studies, posoleucel was well tolerated, with no instances of cytokine release syndrome or other infusion-related toxicities. Six weeks after the first posoleucel infusion, the overall response rate was 95%, and there was a median plasma viral load reduction of 97% [59]. However, in December 2023, the company made the determination to stop its phase III trial due to a review of the data that suggested that the study was unlikely to meet its primary endpoint [60]. Ongoing clinical trials investigating or treating EBV after transplant are listed in Table 2.

Brentuximab vedotin (BV) is a targeted therapy used in the treatment of Epstein–Barr virus (EBV)-associated PTLD, particularly in refractory or relapsed cases. BV combines an anti-CD30 antibody with the cytotoxic agent monomethyl auristatin E (MMAE), delivering this potent drug directly to CD30-expressing lymphoma cells, inducing apoptosis [61,62,63]. Clinical studies, such as the one conducted by Harker-Murray et al. (2023), have shown that BV, alone or in combination with other agents like nivolumab and bendamustine, can lead to significant and durable responses in patients with CD30-positive lymphomas, including PTLD [61,62,63,64]. These findings highlight BV’s effectiveness and manageable safety profile, making it a valuable option, alone or in combination, for heavily pretreated or complex patients suffering from EBV-positive PTLD [61,62,63].

## 7. Future Directions

A major challenge in the management of PTLD includes the establishment of better diagnostic testing that can lead to standardized values for diagnosis. EBV-CTLs show promising results in the therapy of PTLD; however, further studies will be needed to continue to evaluate whether CTLs are efficacious and safe. Additional hurdles include the need to develop new biomarkers for possible therapeutic targets and how to prepare combination therapy and plan for comparative studies [42]. Consideration should also be given to clinical trials investigating novel EBV-CTLs or monoclonal antibodies, small molecule inhibitors, proteasome inhibitors, and new chemotherapy agents [65].

Additional strategies specifically aimed at EBV+ malignancies involve the use of antivirals against EBV. These antiviral approaches under development focus on targeting small molecule inhibitors for EBV-encoded gene products, such as LMP1 and EBNAs [51]. Other proposed methods include inducing the lytic form of EBV replication in tumor cells combined with prodrugs that are cytotoxic to lytically infected cancer cells or enhancing the host immune response to viral antigens expressed by EBV-infected tumor cells [51].

Sirolimus is another potential agent for treating or preventing EBV/PTLD. Research by Peccatori et al. has shown that sirolimus may have a beneficial effect against EBV viremia and PTLD [66]. Sirolimus exhibits inhibitory effects on normal immune system cells and curbs the proliferation of transformed cell lines. Reddy et al. suggested that a combination of low-dose rituximab (100–150 mg/m^2^) and sirolimus, with or without additional immunosuppressive agents, might be the most effective strategy for managing EBV/PTLD. However, this regimen has yet to be reported in use [67].

Recent advances in Epstein–Barr virus (EBV) vaccine development highlight both progress and ongoing challenges. Various vaccine formulations have been explored, including subunit vaccines targeting the gp350 protein, virus-like particles (VLPs), and synthesized mRNA vaccines. The gp350 protein, a major EBV envelope component, has been a primary focus due to its ability to elicit neutralizing antibodies. While phase I and II trials of gp350-based vaccines have shown promise in reducing the incidence of infectious mononucleosis, they have not effectively prevented EBV infection, indicating the need for more comprehensive antigen selection [68,69].

Innovative approaches include combining multiple EBV antigens to enhance immunogenicity, as demonstrated by recent studies on VLPs and mRNA vaccines encoding multiple glycoproteins like gp350, gB, gH/gL, and gp42. Moderna’s phase I clinical trial of an mRNA EBV vaccine has shown encouraging immunogenicity in mice, although concerns about long-term memory responses and potential side effects remain [70]. Efforts also focus on optimizing immunogen design, employing nanoparticle delivery systems, and improving adjuvants to balance cellular and humoral immune responses [68,69]. These advances reflect a concerted effort to overcome the complex lifecycle and infection mechanisms of EBV, aiming for a vaccine that provides robust and long-lasting immunity.

## 8. Conclusions

Epstein–Barr virus (EBV) reactivation is a significant complication in allo-HSCT recipients and may lead to PTLD. Key risk factors include T-cell depletion, use of anti-thymocyte globulin, and certain transplant conditions. In some populations, regular surveillance and preemptive therapy with rituximab are crucial in managing EBV reactivation. Emerging therapies like EBV-specific cytotoxic T lymphocytes show promise for treating refractory cases. Future directions include developing standardized diagnostic tests, new therapeutic targets, and EBV vaccines. Continued research is essential to improve EBV-related adverse outcomes for allo-HSCT patients.

## Figures and Tables

**Table 1 viruses-16-01294-t001:** Risk factors for the development of PTLD after allogeneic transplant.

Factors that Increase the Risk of Development of PTLD	Factors that Decrease the Risk of Development of PTLD
T-cell-depleted transplant	Rituximab use within 6 months pre-HSCT
Use of anti-thymocyte globulin (ATG)	Use of sirolimus for GVHD prophylaxis
Use of alemtuzumab	CD4+ T-lymphocyte count greater than 50 at day 30
Age older than 50	Depletion of both T- and B-cells
Reduced-intensity conditioning	
EBV serology mismatch	
Second HSCT	
Pre-transplant splenectomy	
HLA mismatch	
Haploidentical transplant	
Infusion of mesenchymal stromal cells	
Grade II to IV GVHD	
CMV reactivation	

**Table 2 viruses-16-01294-t002:** Clinical trials (keywords: EBV+ allogeneic, clinicaltrials.gov) investigating the incidence and treatment of EBV reactivation or PTLD after transplant.

Clinical Trial Number	Official Title
NCT06119256	Multicenter, Open Label, Single-arm Exploratory Clinical Study of EBV-TCR-T Cells for EBV Infection After Allogenic HSCT
NCT06391814	Open-Label Individual Patient Study of Epstein–Barr Virus (EBV) Specific T-Cell Lines for the Treatment of a Lymphoproliferative Disease and Hemophagocytic Syndrome Associated With EBV
NCT04554914	An Open-label, Single-arm, Multicohort, Phase 2 Study to Assess the Efficacy and Safety of Tabelecleucel in Subjects with Epstein–Barr Virus-associated Diseases (EBVision)
NCT05532826	Clinical Study on the Efficacy and Safety of Donor EBV-CTL Infusion in Patients with CAEBV and EBV-associated Hemophagocytic Lymphohistiocytosis After Allogeneic Hematopoietic Stem Cell Transplantation
NCT03266653	A Pilot Study in the Treatment of Refractory Epstein–Barr Virus (EBV) Infection with Related Donor EBV Cytotoxic T-Lymphocytes in Children, Adolescents and Young Adult Recipients
NCT04832607	Treatment of Chemo-refractory Viral Infections After Allogeneic Stem Cell Transplantation with Multispecific T Cells Against CMV, EBV and AdV: A Phase III, Prospective, Multicenter Clinical Trial
NCT02580539	A Phase I/II Open-label Study of the Safety and Efficacy of Epstein–Barr Virus Specific T-cell Lines for the Treatment of EBV Infection or EBV-related Lymphoproliferative Diseases
NCT04013802	Administration of Most Closely HLA-matched Multivirus-specific Cytotoxic T-Lymphocytes for the Treatment of EBV, CMV, Adenovirus, and BK Virus Infections Post Allogeneic Stem Cell Transplant
NCT05854225	Thiotepa Incorporating TBI/Cy Conditioning Regimen Followed by HSCT for EBV-HLH With Central Nervous System Involvement: a Prospective Single-arm Clinical Study
NCT05471661	Administration of Rapidly Generated Multipathogen-specific T-Lymphocytes for the Treatment of AdV, CMV, EBV, BKV and Aspergillus Fumigatus Infections Post Allogeneic Stem Cell Transplant
NCT03394365	Multicenter, Open-Label, Phase 3 Study of Tabelecleucel for Solid Organ or Allogeneic Hematopoietic Cell Transplant Subjects with Epstein–Barr Virus-Associated Post-Transplant Lymphoproliferative Disease After Failure of Rituximab or Rituximab and Chemotherapy
NCT02007356	A Phase I/II Single-center Study to Assess Safety and Feasibility of Direct Infusions of Donor-derived Virus-specific T-cells in Recipients of Hematopoietic Stem Cell Transplantation with Post-transplant Viral Infections Using the Cytokine Capture System^®^
NCT02048332	Donor-Derived Viral Specific T-cells (VSTs) for Treatment of Viral Infections After Allogeneic Stem Cell Transplant
NCT06256484	A Phase 1 Study to Evaluate the Safety and Preliminary Efficacy of ATA3219, Allogeneic Anti-CD19 Chimeric Antigen Receptor T-cell Therapy, in Subjects with Relapsed/Refractory B-cell Non-Hodgkin Lymphoma
NCT04288726	A Phase 1 Study Evaluating the Safety and Activity of Allogeneic CD30 Chimeric Antigen Receptor Epstein–Barr Virus-Specific T Lymphocytes (CD30.CAR-EBVSTs) in Patients with Relapsed or Refractory CD30-Positive Lymphomas
NCT05183490	Phase I Study of Adoptive Immunotherapy of Refractory Viral Infection with Ex Vivo Expanded Rapidly Generated Virus Specific T (R-MVST) Cells
NCT03314974	Myeloablative Allogeneic Hematopoietic Cell Transplantation Using a Related or Unrelated Donor for the Treatment of Hematological Diseases
NCT05236764	T-Cell Receptor Alpha Beta+/CD19+ Depletion in Haploidentical Allogeneic Hematopoietic Cell Transplantation (Allo-HCT) for Adult and Pediatric Patients with Hematological Malignancies and Non-malignant Disorders
NCT05805605	Allogeneic Hematopoietic Stem Cell Transplantation Using Reduced Intensity Conditioning (RIC) With Post-Transplant Cytoxan (PTCy) for the Treatment of Hematological Diseases
NCT04195633	Hematopoietic Stem Cell Transplantation from Haploidentical Donors in Patients with Hematological Malignancies Using a Treosulfan-Based Preparative Regimen

## Data Availability

No new data were created or analyzed in this study. Data sharing is not applicable to this article.

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
