# Peer review of "EBV Reactivation and Disease in Allogeneic Hematopoietic Stem Cell Transplant (HSCT) Recipients and Its Impact on HSCT Outcomes"

_viruses, 2024, doi:10.3390/v16081294_

Round 1
Reviewer 1 Report
Comments and Suggestions for Authors
MAJOR
-Needs a comment on the type of PTLD in outcomes: is reactive hyperplasia the same as DLBLC? Or plasmablastic lymphoma?
- The authors comment on the risk of dying with EBV Reactivation vs PTLD. Can the authors address the mechanisms of death in pts with EBV reactivation vs EBV+ PTLD?
MINOR
1. “Risk factors”: line 63: Evaluate what is the correct word: Formation vs formulation
3. “The role of Surveillance”: Line 103. “ in patients treated with Ruxolitinib (30)”:
Can the authors explain the significance of ruxolitinib in the discussion? Or is this a typo for “Rituximab”
Author Response
1) Needs a comment on the type of PTLD in outcomes: is reactive hyperplasia the same as DLBLC? Or plasmablastic lymphoma?
Response: We appreciate the reviewer’s insightful comment. We have included a clarification on the different subtypes of PTLD and their distinctions in the revised manuscript.
Revised Manuscript Section:
“EBV-associated PTLD spans a spectrum of subtypes from benign polyclonal proliferations to malignant monoclonal lymphomas. PTLD includes several subtypes with varying outcomes, including reactive hyperplasia, diffuse large B-cell lymphoma (DLBCL), and plasmablastic lymphoma. Reactive hyperplasia is a benign condition, characterized by the proliferation of normal lymphoid tissue in response to an antigenic stimulus, often linked to EBV, typically with a favorable prognosis. It is an early lesion in PTLD and usually does not progress to malignancy but may resolve with reduced immunosuppression.
Polymorphic PTLD represents a transitional phase with polymorphic cellular infiltrate, architectural distortion of lymphoid tissue, and presence of EBV in most cases. These lesions show clonal or oligoclonal lymphoid populations but do not meet criteria for a specific lymphoma type. It involves malignant, monoclonal B-cell proliferation. Prognosis varies based on the stage at diagnosis, response to immunosuppression reduction, and treatment efficacy. Intensive treatment is often required, with outcomes ranging from favorable to poor.
Monomorphic PTLD includes B-cell neoplasms such asdiffuse large B-cell lymphoma (DLBCL), Burkitt lymphoma, and plasma cell myeloma. These are monoclonal, aggressive, and resemble their counterparts in immunocompetent hosts. T/NK-Cell Neoplasms are less common and include peripheral T-cell lymphoma and hepatosplenic T-cell lymphoma. These lesions have a poorer prognosis compared to B-cell PTLD.
Plasmablastic Lymphoma is a rare, highly aggressive subtype of PTLD, often in-volving the oral cavity and strongly associated with EBV. It has a poor prognosis due to its aggressive behavior and resistance to conventional treatments. Some patients may re-spond to high-intensity chemotherapy. Early diagnosis and aggressive treatment are crucial for improving outcomes.
Classical Hodgkin Lymphoma-type PTLD shares histological features with classical Hodgkin lymphoma and is usually EBV-positive. This type has a better prognosis with appropriate treatment.”
2) The authors comment on the risk of dying with EBV Reactivation vs PTLD. Can the authors address the mechanisms of death in pts with EBV reactivation vs EBV+ PTLD?
Response: We appreciate the author’s request for further clarification on the mechanisms of death in patients with EBV reactivation versus EBV+ PTLD. We have included a detailed explanation in the revised manuscript.
Revised Manuscript Section:
"In patients with EBV reactivation, the primary mechanisms of death include severe immune dysregulation such as cytokine storms or Hemophagocytic Lymphohistiocytosis (HLH), leading to systemic inflammation and multi-organ failure. Direct viral effects, including significant tissue damage and organ invasion, also contribute to mortality. In contrast, patients with EBV-positive Post-Transplant Lymphoproliferative Disorder (PTLD) primarily die from uncontrolled proliferation of EBV-infected B-cells, resulting in extensive organ infiltration and dysfunction. The immunosuppressive therapy required for transplant patients exacerbates this by hindering the immune response, allowing unchecked lymphoproliferation, and leading to complications such as cytokine release syndrome and severe infections." “
3) Risk factors”: line 63: Evaluate what is the correct word: Formation vs formulation
Response: We have changed the word to formulation
“The role of Surveillance”: Line 103. “ in patients treated with Ruxolitinib (30)”:
4) Can the authors explain the significance of ruxolitinib in the discussion? Or is this a typo for “Rituximab”
Response: We appreciate the reviewer’s observation. In the context of our manuscript, ruxolitinib is mentioned deliberately.
Revised manuscript section
“Studies have documented cases of high-grade B-cell lymphomas developing in patients treated with ruxolitinib post-transplant, highlighting the need for careful monitoring and risk assessment in this population”

Reviewer 2 Report
Comments and Suggestions for Authors
I believe you should include a section addressing the spectrum of EBV-PTLD: from polymorphic to monomorphic to lymphoma, as well as the need for diagnostic imaging (CT, CT-PET) and biopsy.
Brentuximab should be mentioned somewhere.
The importance of using always the same source to compare EBV viral load must be emphasized (i.e., whole blood to whole blood, plasma to plasma). When giving copy numbers that trigger action, you must specify if these are whole blood or plasma.
The discussion on EBV vaccines is too superficial and out of date.
Comments on the Quality of English LanguageSome minor editing would be appropriate
Author Response
Reviewer 2
I believe you should include a section addressing the spectrum of EBV-PTLD: from polymorphic to monomorphic to lymphoma, as well as the need for diagnostic imaging (CT, CT-PET) and biopsy.
Response: We appreciate the reviewer's valuable suggestion. We have included a new section in the manuscript addressing the spectrum of EBV-PTLD and the significance of diagnostic imaging and biopsy.
Revised manuscript section:
“EBV-associated PTLD spans a spectrum of subtypes from benign polyclonal prolif-erations to malignant monoclonal lymphomas. PTLD includes several subtypes with varying outcomes, including reactive hyperplasia, diffuse large B-cell lymphoma (DLBCL), and plasmablastic lymphoma. Reactive hyperplasia is a benign condition, characterized by the proliferation of normal lymphoid tissue in response to an antigenic stimulus, often linked to EBV, typically with a favorable prognosis. It is an early lesion in PTLD and usually does not progress to malignancy but may resolve with reduced immunosuppression.
Polymorphic PTLD represents a transitional phase with polymorphic cellular infil-trate, architectural distortion of lymphoid tissue, and presence of EBV in most cases. These lesions show clonal or oligoclonal lymphoid populations but do not meet criteria for a specific lymphoma type. It involves malignant, monoclonal B-cell prolifera-tion. Prognosis varies based on the stage at diagnosis, response to immunosuppression reduction, and treatment efficacy. Intensive treatment is often required, with outcomes ranging from favorable to poor.
Monomorphic PTLD includes B-cell neoplasms such asdiffuse large B-cell lymphoma (DLBCL), Burkitt lymphoma, and plasma cell myeloma. These are monoclonal, aggres-sive, and resemble their counterparts in immunocompetent hosts. T/NK-Cell Neoplasms are less common and include peripheral T-cell lymphoma and hepatosplenic T-cell lymphoma. These lesions have a poorer prognosis compared to B-cell PTLD.
Plasmablastic Lymphoma is a rare, highly aggressive subtype of PTLD, often in-volving the oral cavity and strongly associated with EBV. It has a poor prognosis due to its aggressive behavior and resistance to conventional treatments. Some patients may re-spond to high-intensity chemotherapy. Early diagnosis and aggressive treatment are crucial for improving outcomes.
Classical Hodgkin Lymphoma-type PTLD shares histological features with classical Hodgkin lymphoma and is usually EBV-positive. This type has a better prognosis with appropriate treatment.
Imaging modalities such as CT and PET-CT Scans are crucial in the initial staging and response assessment of PTLD. CT scans help identify the extent of disease, while PET-CT scans provide functional imaging that can distinguish between active disease and fibrotic tissue. PET-CT is particularly valuable in assessing metabolic activity of lesions, guiding biopsies, and evaluating treatment response.
A biopsy is essential for diagnosing PTLD. It allows for histopathological examina-tion, immunophenotyping, and molecular studies to classify the PTLD subtype accu-rately. The biopsy should be sufficiently large to provide material for all necessary tests, including EBV-encoded RNA (EBER) in situ hybridization, immunohistochemistry, and clonality studies”
Brentuximab should be mentioned somewhere.
Response: We appreciate the reviewer's suggestion to include Brentuximab in the discussion. Brentuximab Vedotin (BV) is an important therapeutic option, especially for cases of PTLD that are refractory or relapsed. We have added a section to the manuscript discussing the role of Brentuximab in the treatment of EBV-PTLD.
Revised Manuscript Section
“Brentuximab vedotin (BV) is a targeted therapy used in the treatment of Epstein-Barr Virus (EBV)-associated PTLD, particularly in refractory or relapsed cases. BV combines an anti-CD30 antibody with the cytotoxic agent monomethyl auristatin E (MMAE), delivering this potent drug directly to CD30-expressing lymphoma cells, inducing apoptosis. Clinical studies, such as one conducted by Harker-Murray et al. (2023), have shown that BV, alone or in combination with other agents like nivolumab and bendamustine, can lead to significant and durable responses in patients with CD30-positive lymphomas, including PTLD. These findings highlight BV's effectiveness and manageable safety profile, making it a valuable option, alone or in combination, for heavily pretreated or complex patients suffering from EBV-positive PTLD.
The importance of using always the same source to compare EBV viral load must be emphasized (i.e., whole blood to whole blood, plasma to plasma). When giving copy numbers that trigger action, you must specify if these are whole blood or plasma.”
Response: We appreciate the reviewer’s insightful comment and have incorporated a revised section in the manuscript to emphasize the importance of using consistent sources when comparing EBV viral loads. Additionally, we have specified the type of sample when providing copy numbers that trigger clinical actions.
However, numerous studies have defined EBV thresholds for preemptive treatment as 1000 copies/mL and 10,000 copies/mL in serum or plasma samples, and 40,000 copies/mL in whole blood
The discussion on EBV vaccines is too superficial and out of date.
Response: We appreciate the reviewer’s feedback and have updated the discussion on EBV vaccines to include more comprehensive and current information.
Revised Manuscript Section
“Recent advances in Epstein-Barr virus (EBV) vaccine development highlight both progress and ongoing challenges. Various vaccine formulations have been explored, including subunit vaccines targeting the gp350 protein, virus-like particles (VLPs), and synthesized mRNA vaccines . The gp350 protein, a major EBV envelope component, has been a primary focus due to its ability to elicit neutralizing antibodies. While phase I and II trials of gp350-based vaccines have shown promise in reducing the incidence of infectious mononucleosis, they have not effectively prevented EBV infection, indicating the need for more comprehensive antigen selection.

Reviewer 3 Report
Comments and Suggestions for Authors
Well written, pretty complete and up to date; useful for scientists and clinicians
Author Response
Response: We appreciate the positive feedback. Thank you!
Round 2
Reviewer 2 Report
Comments and Suggestions for Authors
Thank you for addressing my comments